# Prevalence and Predictors of Food Insecurity among Adults with Type 1 Diabetes: Observational Findings from the 2022 Behavioral Risk Factor Surveillance System

**DOI:** 10.3390/nu16152406

**Published:** 2024-07-25

**Authors:** Julie Ann Wagner, Angela Bermúdez-Millán, Richard S. Feinn

**Affiliations:** 1Division of Behavioral Sciences and Community Health, School of Dental Medicine, UConn Health, 263 Farmington Avenue, Farmington, CT 06030-3910, USA; 2Department of Public Health Sciences, School of Medicine, UConn Health, 263 Farmington Avenue, Farmington, CT 06030-6325, USA; bermudez-millan@uchc.edu; 3Department of Medical Sciences, Quinnipiac University, Hamden, CT 06518, USA; richard.feinn@quinnipiac.edu

**Keywords:** type 1 diabetes, food insecurity, Behavioral Risk Factor Surveillance Survey

## Abstract

The majority of data on food insecurity in diabetes comes from samples of type 2 diabetes or youth with type 1 diabetes. This study screened for food insecurity among adults with type 1 diabetes in the 2022 Behavioral Risk Factor Surveillance Survey, which was the first year that respondents who endorsed diabetes were asked to indicate whether they had type 1 or type 2. One validated screening item asked, “During the past 12 months, how often did the food that you bought not last and you didn’t have money to buy more?”. Respondents who answered “always”, “usually”, “sometimes”, or “rarely” were categorized as having a positive screen for food insecurity. Seventy-six percent of the sample was white/non-Hispanic. Over one-quarter screened positive for food insecurity. This prevalence is higher than some reports of food insecurity in type 1 diabetes but consistent with reports that include ‘marginal’ food security in the count of food-insecure individuals. White/non-Hispanics had a lower risk of a positive screen than minoritized respondents. Respondents reporting older age, lower educational attainment, not working, lower income, and receiving SNAP benefits had higher rates of a positive screen. Significant healthcare factors associated with a positive screen were receiving government insurance instead of private, not being able to afford to see a doctor, and worse general, physical, and mental health. In conclusion, rates of a positive screen for food insecurity among people with type 1 diabetes in this study were alarmingly high and associated with other socioeconomic indicators. Screening for food insecurity with appropriate instruments for samples with type 1 diabetes, across the U.S. and internationally, should be a priority.

## 1. Introduction

About 2 million Americans have type 1 diabetes (T1D), and the vast majority (85%) are adults [1]. Healthy eating with careful attention to carbohydrates is a cornerstone of T1D self-management. Food insecurity (FI) is the limited or uncertain availability of nutritionally adequate and safe foods, or the limited or uncertain ability to acquire acceptable foods in socially acceptable ways [2]. FI in T1D has been linked to hyperglycemia. In one study, youth with T1D from FI households had 2.37-higher odds of high-risk glycemic control (HbA1c > 9.0%) and 2.95 times the prevalence rate of emergency department visits compared to their counterparts from food-secure households [3]. FI is also associated with an increased risk for acute life-threatening complications including 1.58 times the odds of an episode of diabetic ketoacidosis and 1.53 times the odds of severe hypoglycemia [4,5]. Compared to their food-secure counterparts with T1D, FI individuals with T1D also report more diabetes stigma [6] and more fear of hypoglycemia [7].

Most reports regarding FI in T1D come from youth and young adults with newly diagnosed T1D, with rates of FI ranging from 19.5% [3] to 16.4% [4]. The Behavioral Risk Factor Surveillance System (BRFSS) is a U.S. health survey coordinated by the national Centers for Disease Control and Prevention and conducted by participating individual state health departments. Beginning in 1984, more than 350,000 adults are interviewed each year. The 2022 BRFSS was the first year that BRFSS asked respondents who endorsed having diabetes to distinguish whether they had been diagnosed with T1D or T2D, and whether they were taking insulin. This affords the first opportunity to examine rates of FI in a representative sample of U.S. adults with T1D. We sought to examine the prevalence and predictors of a positive screen for FI among adults with T1D. 

## 2. Materials and Methods

The BRFSS design uses state-level, random digit dialed probability samples of adults (ages 18 and older). The questionnaire consists of 3 parts: (1) a core component of questions used by all states, which includes questions on demographics and current health-related conditions and behaviors; (2) optional CDC modules on specific topics (e.g., diabetes) that states may elect to use; and (3) state-added questions, developed by states for their own use. Interviews are generally conducted using computer-assisted telephone interviewing (CATI) systems. Data are weighted for noncoverage and nonresponse.

The 2022 BRFSS dataset includes landline and cell phone data. The optional Diabetes module was administered in California, Delaware, Georgia, Indiana, Maine, Maryland, Massachusetts, Mississippi, Missouri, Nevada, New York, North Dakota, Ohio, South Carolina, South Dakota, Vermont, Virgin Islands, and Wisconsin [8]. These regions use a disproportionate stratified sample design except for Virgin Islands which used a simple random sample design. Participants who endorsed T1D and who reported taking insulin were considered to have T1D and were included in analyses.

### 2.1. Measures

BRFSS screened for FI using a validated screening item [9], “During the past 12 months, how often did the food that you bought not last and you didn’t have money to buy more?” [10]. Response options were “never”, “rarely”, “sometimes”, “usually”, or “always”. People with ‘marginal food security’, usually classified as food secure in the U.S. government’s prevalence estimates, may also face impaired health and nutrition [11]. Therefore, consistent with Bickel [10], affirmative responses to the item were classified as being a positive screen for FI. Participants who responded “never” were the reference group.

Demographics included sex, age, education, count of children in the household, employment, income, insurance, and SNAP benefits (yes vs. no). Due to small cell counts, race and ethnicity were dichotomized as white/non-Hispanic vs. minoritized. 

Health and healthcare characteristics included “have a personal doctor” (yes vs. no), “couldn’t afford to see a doctor” in the past 12 months (yes vs. no), self-reported health (excellent to poor), count of days that physical health was “not good” and mental health was “not good” in the past month, and body mass index (BMI) calculated from self-reported height and weight.

### 2.2. Statistical Analysis

Complex sample analyses accounted for the BRFSS complex survey design. Logistic regression tested for associations between respondent characteristics and a positive screen for FI. A multivariable backward stepwise logistic model determined the variables that predict a positive screen for FI. Analyses using ordinal regression on the original 5-level FI question resulted in similar results and are not presented due to space limitations. SPSS v29 was used with the statistical significance set at 0.05. 

## 3. Results

The BRFSS 2022 database contains 626 respondents reporting having T1D and taking insulin. Of them, 469 (74.9%) answered the FI question and were included in analyses. The response distribution for the FI question was n = 15 (3.4%) always, n = 5 (1.0%) usually, n = 47 (11.4%) sometimes, n = 50 (10.6%) rarely, and n = 352 (73.7%) never. The majority were white (77.0%) or Black (11.1%), and the remainder were American Indian/Alaska Native (0.8%), Asian (0.2%), or multiracial (6.9%); 5.9% were Hispanic. The reference group white/non-Hispanic was 76.0%. 

Table 1 shows the demographic characteristics and the bivariate logistic regression associations between the respondent characteristics and a positive screen for FI. The significant associations were race/ethnicity, where minoritized respondents were more likely to have a positive screen than white/non-Hispanics; education, where lower educational levels had higher rates of a positive screen; and working status, income, and SNAP benefits, where not working, having a lower income, and receiving SNAP benefits were associated with greater positive screens for FI. The significant healthcare factors associated with FI were receiving government insurance instead of private, not being able to afford to see a doctor, and worse general, physical, and mental health. Figure 1 displays which factors are positively and negatively associated with a positive screen for FI. 

Associations with a Positive Screen for Food Insecurity in Adults with Type 1 Diabetes.

The results from the stepwise multivariate model are shown in Table 2. Five variables were statistically significant, and one was age, which was not significant in the bivariate analysis. Adjusting for the other variables in the model, an increase in age was associated with a lower odds of a positive screen for FI. The remaining variables in the multivariate model were consistent with the bivariate analyses. Overall, the model explained a decent proportion of the variability in FI (pseudo R^2^ = 0.45) and correctly classified 72% of the sample.

## 4. Discussion

In this study, over one-quarter of participants with T1D screened positive for FI. This is higher than some previous reports of FI ranging from 19.5% [3] to 16.4% [4]. The BRFSS rates may be higher for several reasons. First, BRFSS assessed FI with a single question while other studies used multi-item scales with differing cutoffs to determine FI. The relatively more liberal criterion reported here was a single item and included “rarely” responses, which is judged to be an important inclusion in light of the fact that even ‘marginal’ FI conveys nutritional and health risks [11]. A recent report on T1D [4] found 7.8% low food security, 8.6% very low food security, and 11.7% marginal food security, totaling 28.1%—which is in fact not dissimilar to our finding of 26.3%. Second, the BRFSS sample includes adults rather than youth. Data show that in FI households, feeding youth is often prioritized over feeding adults [12], such that youth may have less experience running out of food than adults do. Third, clinic-based [3] or internet-based [5] studies may bias respondent characteristics toward healthier individuals with a lower exposure to FI. 

Additionally, some scales likely underestimate FI [13]. One study of people with either type 1 or type 2 diabetes concluded that food security scales may underestimate FI because of response patterns in people with diabetes and possibly attributes of the disease itself and its acute complications [13]. That study suggests that traditional screening questions and cutoffs could miss FI among a substantial proportion of individuals with diabetes, with their study finding 5% to 8% erroneously categorized as food secure. The authors warn that the standard measurement of FI among adults, particularly those with chronic illness, could be problematic for estimations of national food security. 

FI in T1D may lead to ‘eat-or-treat’ decisions, i.e., the decision whether to spend limited money on food or insulin. In fact, recent data show that rationing insulin is not uncommon. In an international study, the U.S. had the highest percentage of participants rationing insulin over the previous year, and they rationed most frequently, i.e., weekly or more often. Insulin deficiency causes hyperglycemia and can lead to diabetic ketoacidosis within hours. Survival in absolute insulin sufficiency is only days-to-weeks. Insufficient food, on the other hand, can lead to dangerously low blood glucose levels and episodes of severe hypoglycemia which can also be life-threatening. Thus, eat-or-treat decisions in T1D can have serious consequences. While we did not measure ‘eat-or-treat’ decisions, our findings do show that respondents endorsing FI had lower self-rated health and had more “not good” mental and physical days.

Our U.S. findings may not generalize to other countries, but internationally, difficulties managing T1D in the setting of FI have been documented in developed countries [14], low- and middle-income countries [15,16,17], and in conflict settings [18]. It can be reasonably expected that ‘eat-or-treat’ decisions may be at least as common in low-resource settings, where food deprivation is not uncommon and food assistance programs are meager, as they are in higher-resource settings with safety net programs such as the U.S. 

In the multivariate analysis, younger adults had increased odds of a positive screen for FI compared to older adults. Young adults are particularly vulnerable to the effects of stressors on diabetes control [19]. The results suggest the need for FI screening among people with T1D, especially young adults. Because these data also show that individuals at-risk for FI may not attend routine clinic appointments due to cost, screening in emergency departments should be considered, where patients present with acute complications such as DKA and severe hypoglycemia. 

### 4.1. Limitations 

Some individuals with T2D might have erroneously endorsed having T1D. This is mitigated to some extent by a “don’t know” response option which reduces guessing and by making insulin use an inclusion criterion. The BRFSS did not administer a multi-item food security scale. However, the single item that was administered references food actually running out, which is a relatively stringent screening item. The BRFSS survey relies exclusively on self-report data which can be prone to errors due to biases such as social desirability, cognitive processes such as forgetting, survey conditions such as time constraints for answering questions, and characteristics of the survey itself such as directive questions. The BRFSS survey data analyzed here were limited to the several states in the U.S. that administered the optional Diabetes module. Studies are warranted across all 50 states and five inhabited territories, including especially the most populous territory, Puerto Rico, where approximately 1/3 of the population is food insecure [20] and rates of T1D are rising steeply [21]. International studies on the prevalence of FI in T1D are also sorely needed.

### 4.2. Conclusions

Rates of FI are alarmingly high among adults with T1D in the U.S. The COVID-related increases in nutrition assistance programs are being rolled back, potentially increasing FI in those households that depend on such assistance for food. Given the numerous clinical and psychosocial diabetes outcomes that are worsened by FI, screening for FI in T1D should be a priority. International research, particularly in low- and middle-income countries, should examine the site-specific rates and consequences of FI in T1D. Policy decisions about food assistance should consider the implications of including chronic illness in eligibility criteria. 

## Figures and Tables

**Figure 1 nutrients-16-02406-f001:**
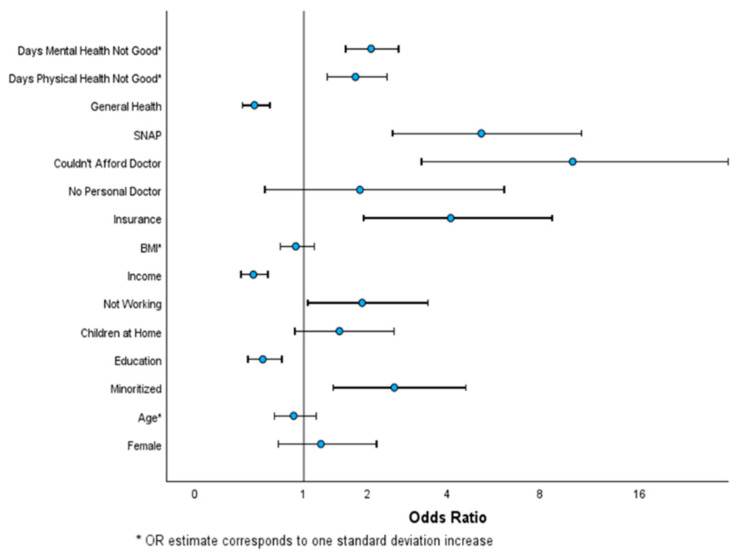
Odds ratio with 95% CI of bivariate associations with a positive screen for food insecurity.

**Table 1 nutrients-16-02406-t001:** Demographic characteristics and bivariate association with a positive screen for food insecurity in adults with type 1 diabetes.

Demographic	Unweighted Count	Weighted Percentage	Percentage with Positive Screen for FI	Odds Ratio	95% CI	*p*-Value
*Sex*						
Female	226	45.3	28.5	1.23	0.70–2.18	0.473
Male	243	54.7	24.5	reference		
*Age*						
Mean ± SE, Coefficient B		49.4 ± 1.3	B = −0.008	0.99	0.98–1.01	0.360
*Race*						
Minoritized *	88	26.0	44.1	2.56	1.41–4.62	0.002
White/Non-Hispanic	364	74.0	21.8	reference		
*Education*						
Less than high school	38	12.7	49.1	8.05	2.85–22.71	<0.001
High school	115	26.2	32.0	3.93	1.77–8.72	
Attended college	136	33.4	26.4	2.99	1.18–7.59	
Graduated college	178	27.1	10.7	reference		
DK/Refused	2	0.6				
Linear Predictor Coefficient B ^			B = −0.615	0.54	0.40–0.74	<0.001
*Children in Household*						
Children	99	27.2	32.4	1.51	0.89–2.56	0.125
No children	368	72.8	24.1	reference		
*Employment*						
Not working	286	52.4	32.3	1.90	1.05–3.42	0.033
Working	181	46.9	20.1	reference		
Refused	2	0.7				
*Income*						
<25 k	96	20.6	46.7	23.78	2.95–191.53	<0.001
25–50 k	98	17.4	43.4	20.81	2.55–169.9	
50–100 k	127	28.3	23.2	8.19	0.90–74.37	
100–200 k	57	13.4	0.5	0.14	0.0–74.37	
>200 k	18	5.8	3.6	reference		
DK/Refused	73	14.5				
Linear Predictor Coefficient B ^			B = −0.797	0.45	0.34–0.59	<0.001
*Health Insurance*						
Government	261	45.8	40.0	4.10	1.93–8.73	0.001
None	10	2.7	28.5	2.45	0.30–20.41	
Private	188	47.8	14.0	reference		
DK/Refused	10	3.6				
*Received SNAP*						
Yes	57	11.2	59.9	5.20	2.52–10.73	<0.001
No	408	88.0	22.3	reference		
DK/Refused	4	0.8				
*Have Personal Doctor*						
No	17	4.5	39.2	1.86	0.56–6.17	0.313
Yes	449	95.1	25.8	reference		
DK/Refused	3	0.3				
*Couldn’t Afford to See Doctor*						
Yes	35	9.7	73.1	10.10	3.23–31.63	<0.001
No	431	90.2	21.2	reference		
DK/Refused	3	0.1				
*General Health*						
Poor	65	13.6	56.3	16.78	2.77–101.83	<0.001
Fair	117	25.4	31.7	6.05	1.03–35.41	
Good	177	38.6	22.1	3.68	0.58–23.57	
Very good	95	19.3	8.3	1.18	0.17–8.33	
Excellent	14	2.9	7.1	reference		
DK/Refused	1	0.3				<0.001
Linear Predictor Coefficient B ^			B = −0.779	0.46	0.35–0.61	
*Days Physical Health Not Good*						
Mean ± SE, Coefficient B		7.9 ± 0.7	B = 0.046	1.05	1.01–1.08	0.008
*Days Mental Health Not Good*						
Mean ± SE, Coefficient B		6.6 ± 0.6	B = 0.069	1.07	1.04–1.11	<0.001
*BMI*						
Underweight	7	1.3	35.2	1.21	0.17–8.32	0.599
Normal	134	31.2	31	reference		
Overweight	156	35.1	22.3	0.64	0.32–1.28	
Obese	156	32.3	25.8	0.77	0.39–1.55	
Mean ± SE		29.2 ± 0.6				
Linear Predictor Coefficient B ^			B = −0.139	0.870	0.62–1.22	0.413

* Minoritized—combines Black (59), American Indian/Alaska Native (5), Asian (1), multiracial (10), and Hispanic (16). ^ Linear predictor—ordinal variable modeled as a quantitative continuous variable. SNAP = Supplemental Nutritional Assistance Program. DK = Don’t Know.

**Table 2 nutrients-16-02406-t002:** Backward stepwise multivariate logistic regression predicting a positive screen for food insecurity in adults with type 1 diabetes.

Demographic	Odds Ratio	95% CI	*p*-Value
*Age*			
One year increase	0.98	0.95–1.00	0.019
*Income*			
One category increase	0.57	0.43–0.77	<0.001
*Health Insurance*			
Government	5.25	2.07–13.30	0.003
None	3.34	0.37–30.39	
Private	reference		
*Couldn’t Afford to See Doctor*			
Yes	11.21	3.11–40.42	<0.001
No	reference		
*Days Mental Health Not Good*			
One day increase	1.05	1.02–1.09	<0.001
*Fit Statistics*			
Nagelkerke R-square	0.447		
Correct Classification			
Food Secure	86.9%		
Food Insecure	34.0%		
Overall	72.0%		

## Data Availability

Data are available at: https://www.cdc.gov/brfss/annual_data/annual_2022.html

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
