# Peer review of "Prevalence and Predictors of Food Insecurity among Adults with Type 1 Diabetes: Observational Findings from the 2022 Behavioral Risk Factor Surveillance System"

_nutrients, 2024, doi:10.3390/nu16152406_

Round 1
Reviewer 1 Report
Comments and Suggestions for Authors
I suggest the authors rewrite the abstract, first, they should provide a background statement, and only then, the objectives of the study should be mentioned; Abbreviations should be avoided in this section; What can we conclude from this report? Future perspectives and directions for further investigations?
“adult” is not an adequate keyword for this manuscript.
Format your references according to the journal’s guidelines.
You should structure your manuscript in 1) Introduction; Materials and Methods; Results and Discussion; and Conclusion. See some examples:
https://www.mdpi.com/2072-6643/16/13/2022
https://www.mdpi.com/2072-6643/16/11/1621
https://www.mdpi.com/2072-6643/16/10/1527
The objective section is not adequate. Please, see the examples I mentioned before.
Considering the obtained results, they should be better discussed with more studies around the world. Please, improve and expand your Discussion section.
“Limitations” should be a subsection presented at the end of Discussion section and please, also elaborate more on your Conclusions in an individual section and provide some directions for future studies and the practical implications of the results you obtained.
Author Response
Comment 1: I suggest the authors rewrite the abstract, first, they should provide a background statement, and only then, the objectives of the study should be mentioned; Abbreviations should be avoided in this section; What can we conclude from this report? Future perspectives and directions for further investigations?
Response 1: The abstract has been revised according to the recommendations above.
Comment 2: “adult” is not an adequate keyword for this manuscript.
Response 2: This is an important comment. The term adult is removed from the list of keywords, and appears in the title instead. We judge that, because the preponderance of research in Type 1 diabetes focuses on children, our focus on adults (who comprise the majority of individuals with Type 1) should be highlighted.
Comment 3: Format your references according to the journal’s guidelines.
Response 3: Thank you, we have re-formatted the references.
Comment 4: You should structure your manuscript in 1) Introduction; Materials and Methods; Results and Discussion; and Conclusion. See some examples:
https://www.mdpi.com/2072-6643/16/13/2022
https://www.mdpi.com/2072-6643/16/11/1621
https://www.mdpi.com/2072-6643/16/10/1527
Response 4: Thank you for suggesting the examples. Accordingly, we have structured the manuscript with the headings suggested by the reviewer.
Comment 5: The objective section is not adequate. Please, see the examples I mentioned before.
Response 5: We note that the reviewer rated the introduction as “must be improved” and have modified it accordingly. We have renamed the “objective” section as “introduction” and provide a more thorough literature review.
Comment 6: Considering the obtained results, they should be better discussed with more studies around the world. Please, improve and expand your Discussion section.
Response 6: We have added international literature to the discussion section and recommended that international studies on the topic are needed.
Comment 7: “Limitations” should be a subsection presented at the end of Discussion section
Response 7: There is a “limitations” subsection at the end of the “discussion” section.
Comment 8: and please, also elaborate more on your Conclusions in an individual section and provide some directions for future studies and the practical implications of the results you obtained.
Response 8: We note that the reviewer rated the conclusions as “must be improved”. In response, we have added a “conclusions” section in which we provide suggestions for future studies and practical implications of our findings.
Reviewer 2 Report
Comments and Suggestions for Authors
Dear Author, thank you for sharing this research.
Please add in the title the type of the study.
Regarding your manuscript:
The main question addressed by the research are the prevalence and predictors of food insecurity risk in adults with T1D.
The study is relevant because addresses the gap in understanding how FI affects adults with established T1D. This study adds to the field by using data from the 2022 BRFSS, which for the first time distinguished between T1D and T2D. Additionally, the study identifies specific sociodemographic and health factors associated with FI risk in this population.
The study relies on self-reported data, which can be prone to bias. It should be explain in the text. However, Including objective measures of food insecurity could strengthen the findings.
In conclusion, this study successfully identified significant predictors of FI risk in adults with T1D using logistic regression analysis. The findings support the argument that social determinants of health, such as income and access to healthcare, play a role in FI risk for this population.
Author Response
Comment 1: Please add in the title the type of the study.
Response 1: We have added to the title the terms “prevalence and predictors” and “observational”
Comment 2: The main question addressed by the research are the prevalence and predictors of food insecurity risk in adults with T1D.
The study is relevant because addresses the gap in understanding how FI affects adults with established T1D. This study adds to the field by using data from the 2022 BRFSS, which for the first time distinguished between T1D and T2D. Additionally, the study identifies specific sociodemographic and health factors associated with FI risk in this population.
The study relies on self-reported data, which can be prone to bias. It should be explain in the text. However, Including objective measures of food insecurity could strengthen the findings.
Response 2: We agree that self-reported data are prone to errors. This is now addressed in the limitations section.
Comment 3: In conclusion, this study successfully identified significant predictors of FI risk in adults with T1D using logistic regression analysis. The findings support the argument that social determinants of health, such as income and access to healthcare, play a role in FI risk for this population.
Response 3: Thank you.
Round 2
Reviewer 1 Report
Comments and Suggestions for Authors
Thank you for following my suggestions.